# Climate Change in the Arctic—The Need for a Broader Gender Perspective in Data Collection

**DOI:** 10.3390/ijerph18020628

**Published:** 2021-01-13

**Authors:** Arja Rautio, Natalia Kukarenko, Lena Maria Nilsson, Birgitta Evengard

**Affiliations:** 1Faculty of Medicine, University of Oulu, FI-90014 Oulu, Finland; arja.rautio@oulu.fi; 2Thule Institute, University of the Arctic, FI-90014 Oulu, Finland; 3Department of Philosophy and Sociology, Northern Arctic Federal University, 163002 Arkhangelsk, Russia; n.kukarenko@narfu.ru; 4Arcum, Arctic Research Centre, Umeå University, 901 87 Umeå, Sweden; lena.nilsson@umu.se; 5Department of Clinical Microbiology, Umeå University, 901 87 Umeå, Sweden

**Keywords:** gender, Arctic, health, policy, human rights, quality, economic benefits

## Abstract

Climate change in the Arctic affects both environmental, animal, and human health, as well as human wellbeing and societal development. Women and men, and girls and boys are affected differently. Sex-disaggregated data collection is increasingly carried out as a routine in human health research and in healthcare analysis. This study involved a literature review and used a case study design to analyze gender differences in the roles and responsibilities of men and women residing in the Arctic. The theoretical background for gender-analysis is here described together with examples from the Russian Arctic and a literature search. We conclude that a broader gender-analysis of sex-disaggregated data followed by actions is a question of human rights and also of economic benefits for societies at large and of the quality of services as in the health care.

## 1. Introduction

Climate change poses a number of significant risks for mankind, with direct and indirect effects on human health. [1,2]. The Arctic is warming at almost three times the global average rate and there is a growing awareness that feedback loops are turning the Arctic into a contributor to climate change. There is a population of more than 4 million people in the Arctic region. This population consists of women, men, girls, and boys and the impacts on these groups due to the rapid progress of climate change differ, as gender related activities differ. During the Second UArctic Congress in Finland, September 2018 a special side-event “Women of the Arctic” was arranged to discuss issues of gender (in)-equality in the Arctic. The discussion at this congressional meeting revealed that women continue to experience significant inequalities, whereas men enjoyed historical advantages, such as leadership positions. One recommendation that came out from this congressional meeting was to introduce policies seeking gender equity, and to include more men in gender equity discussions. It was, however, pointed out that gender equality cannot be achieved within the conferences or side-events called like “Women of the Arctic” because it directly excludes male participation both as audience and speakers. It was expressed, that if the term gender becomes a substitute for women, then we imply that women’s issues are better understood by women and there is no point in inviting men into discussions about women’s issues since men lack experience and expertise in the field. The entanglement of the terms women and gender becomes problematic when gender issues become substituted with women’s issues, as it limits the scope of the discussions to one gender only and consequently women become the target audience and are seen as the only actors. What is more, equalizing gender issues with women’s issues makes women a homogeneous group. Feminists of the postmodern and postcolonial epistemology have criticized this unification since it denies differences between genders and the plurality of their identities, experiences, and practices. Gender equality is about equality between and within diverse groups of women and men.

From gender equality perspective it is not so relevant to only focus on how, e.g., many women and men have different diseases, but to relate the findings to the context in which women and men live. Lifestyles of diverse groups of women and men are impacted by environment, as how and why persistent organic pollutants (POPs), lead, and mercury get into their bodies and how society at large and health institutions react to women’s and men’s health complaints. Women in most Arctic Indigenous communities are generally reported to consume proportionally less traditional food than men, which puts men into higher risks of exposure to heavy metals and POPs predominantly associated with traditional food consumption [3,4]. When the mainstreaming of gender equality into research and policies suddenly is limited to women, we lose the perspective of not only men but also of many women: urban/ rural; Indigenous/non-Indigenous; young and old; pregnant and with small children, etc. This knowledge is of equal importance both to women and men and both sexes should be aware of the processes taking place in their local communities. Sustainable development is possible when we have data on vulnerable groups in the Arctic in order to understand the needs for improvement. Gender is one of the variables, but it has more empowering capacity in comparison to sex as it allows an in-depth analysis of the processes taking place in the Arctic communities.

Statistics is the core of monitoring and for a sustainable development anywhere. Today, there is a general agreement that these statistics should be gender divided as men and women, to a certain extent, have different livelihoods, and thus are exposed to different health hazards. As part of the work in a Nordic center of Excellence, CLINF (https://www.clinf.org), funded by Nordforsk, the gender perspective has been kept in focus of all projects performed within these frames between 2016 and 2021. Here, as part of the outcome of CLINF, the aim is to stress the importance of broadening the gendered statistics as climate change in the Arctic likely affects so far than women and men, and girls and boys differently. However, gender as a variable remains weakly represented in research and policies [5] and a theoretical background is given together with descriptions of currents situations for Indigenous peoples.

## 2. Methods

This study consists of a literature review and a case study. The scoping literature search was performed in 2019 as a reply to a study performed in 2010, using the PubMed and Web of Science databases and the key-words: “climate change”, “human health”, “gender”, and “policies” [5]. In the case study carried out in Russian Arctic zone (manuscript in writing) within the frame of the CLINF project several expeditions have been taken to the remote settlements of the Arkhangelsk region in 2016–2018. The methods used were in brief questionnaires spread among the people living in Nenets Autonomous Okrug, on Kolguev and Vaigach Islands (145 persons) as well as semi-structured interviews with local people (24 persons). The main aim was to find out if people see their future in the current place of residence and how safe and comfortable they feel in the light of climate and environment change. Differences between groups were compared numerically and ordinally.

## 3. Results

### 3.1. Literature Seach

The literature search rendered a similar pattern as in 2010, where the proportion of publications including climate change as well as human health and gender had not changed in a substantial way (Table 1). However, the increasing number of publications on a general level was mirrored by an increasing number of publications taking all keywords into account. A shown in Table 1 articles containing all search words (climate change, health, gender, and policy) increased from 0 to 5 articles found in PubMed and 7 found in Web of Science. So here we see a positive trend, however small, towards a better following of requests of decisionmakers.

### 3.2. Case Study Results

Our respondents consisted of 51% men and 49% women. They were 20–59 years old, mostly having secondary or specialized secondary education. Half of the respondents of both sexes reported having children. While 78% of both sexes responded that they wanted to continue living in their current place of residence, only half of the respondents wanted their children to stay in their community.

Female respondents mostly showed less certainty in their abilities and opportunities to change something or influence the authorities on the issues of health services or ecological and environmental situation. This might partially explain why women want their children to move from the Arkhangelsk region. At the same time observations and interviews showed that women are the main actors in the local activities, doing most of social networking and communication. This is connected to the social–economic context when Indigenous and local men either are unemployed or work in the “traditional” economies like natural resources extraction enterprises, in timber industry or fishing, hunting, and reindeer herding without engaging into domestic work. Unlike men, many women have to combine paid work with care-taking activities for the family [6]. This might explain to some extent why male respondents claim that they keep living in traditional lifestyle while women see the differences between generations and claim living in a different world in comparison to their parents.

In addition, at the community level, we can observe a gender segregation of the labor market. Women mostly work in local administration, education, trade, cultural, or health institutions. Consequently, women are more secure with a regular income and often have more stable and more prestigious social and economic statuses in the remote small communities. Local women are the ones to perform public activities and to engage into new “non-traditional” social and economic projects [7,8].

Answers to questions on climate change and ecology show that both women and men notice the changes, men somewhat more than women probably because they spend more time outdoors. When explaining the sources of information on climate change and pollution the women mainly rely on newspapers (44%) and television (31%) as the main sources of information while men show more trust to internet resources (36%) and their own observations (37%), which we explain by the fact of gendered labor division. Our respondents showed that they generally do not trust the research and official information on the ecological situation and on contaminants. A number of respondents noted that official reports are written in the language that is difficult for non-specialists to understand.

## 4. Discussion

Within the context of this article we find it useful to apply Carol Bacchi’s approach of political deconstruction which is “What’s the problem represented to be?” [9]. Bacchi suggests that each presentation of a problem as a problem already implies certain interpretation of the phenomenon. So the analysis should include—what assumptions underlie this representation, what effects are produced by this representation and what is left unproblematic in it?

The literature search shows a growing, but still small, interest in linking climate change, gender, and health in Arctic research. This is also mirrored in a recent report “Arctic Policies and Strategies—Analysis, Synthesis and Trends”, analysis of the policies, strategies, and declarations of the relevant Arctic stakeholders including also new and/or emerging trends of Arctic governance and based on a coding of the text of 56 policy documents (in 1996–2019) show the same result [10]. Gender analysis is rarely described in the texts and gender equality rarely mentioned. In the texts from one of the countries it is not found at all while mentioned for Greenland in the Danish documents. The perspective is mostly found in Swedish documents and publications [11,12].

The Russian example illustrates how sex-divided statistics is an important dimension of demographic registers and plays a significant role for understanding which demographic characteristics of the people residing in the Arctic along with ethnicity, age, education background, family status, etc., are important for decision-makers in order to formulate policies addressing concrete target groups. Just like any quantitative statistics sex-disaggregated data is an important variable to be monitored in the population registers.

Within the context of the CLINF research project with the word sex we refer to biological differences between persons registered as women and men. Sex characterizes demographic biological attributes that are ascribed to humans at birth. By gender, however, we understand socially constructed behavior patterns, norms, and values that women and men learn during their upbringing, education, and socialization and then interiorize as part of their social character. What is important for us in the project is that gender relations are social, constructed, and include asymmetries not only between women and men but also within the two sexes depending on ideological, historical, cultural, religious, ethnic, and economic background and consequently might vary from society to society and can change under the circumstances [13].

Our research approach departs from the distinction of sex/gender and understands sex as a biological binary and gender as a social construction “rooted in biology and shaped by experience and environment”. Still most of the current studies on climate change impacts on human health use “gender” meaning biological belonging of humans to two sexes and indicate the number of men and women as two categories having some biologically-based male-female differences in reactions to environmental changes and pollution [14,15].

Gender statistics is not equal to sex-divided statistics and implies qualitative analysis of social institutions and relations impact on the health situation of women and men in the Arctic, on their vulnerabilities and opportunities in order to better understand what decision-makers are to do in order to achieve sustainable development. Sex-disaggregated data as it has already been mentioned above is important for developing gender analysis and gender-sensitive policies which are tools needed to tackle different impacts of climate change on peoples’ lives. During the last 25 years, several international bodies and conferences have signed documents stating that the gender perspective should be integrated into policy and other documents concerning activities where humans are involved on an international, national, regional, and local level [16]. Women’s key role in ensuring sustainable development was highlighted in the Brundtland Commission on Environment and Development’s report Our Common Future as did United Nations Conferences like the Beijing Declaration and Platform for Action 1995 and the World Summit on Sustainable Development 2002. The UN Women was established in 2010 and in 2007 WHO adopted a resolution on the integration of gender analysis and action into the work of WHO at all levels.

### Development of Health among Arctic People—An Increased Need of Adequate Data

As in many parts of the world, several health aspects have been improved in the Arctic. Increase in the life expectancy and decrease in infant mortality, as well as reduction in mortality from infectious diseases all count as improved health. The risks of new infectious and zoonotic diseases have increased in the Arctic due to climate warming ([17,18]. At the same time, there are an increase in chronic diseases such as cardiovascular disease, stroke, hypertension, diabetes, and obesity [3,19,20]. Due to rapid social change, dietary transitions, decreased physical activities, and exposure to new environmental hazards in the Arctic, these negative increases happen in particular in traditional societies. In this context, the Sami people differ from other circumpolar Indigenous groups—with a risk pattern of uncommunicable diseases similar to the one of the majority population of the Nordic countries [21,22,23].

It has also been shown that Sami, with a more deviant lifestyle compared to the majority population, i.e., reindeer herding Sami, have a more deviant health pattern compared to the majority population than non-reindeer herding Sami [21,24,25].

Humans are exposed to many environmental factors with a potential impact on their health. Income, education, social status, genetics, and access to social safety networks and welfare, are facts that which all influence individual health. Combined with personal lifestyle choices, coping capacity, and access to good primary health care, the impact of exposure to environmental contaminants may seem small. However, a number of studies document that exposure to contaminants such as mercury and PCBs can have adverse effects on women of childbearing age, fetuses, and small children (AMAP, 2009; 2015). According to AMAP reports, environmental exposure is usually an exposure not only to a single substance, but to a mixture of chemicals in the environment. The trends of most “traditional” contaminants have decreasing levels in blood samples of pregnant women; however, new ones, which may have endocrine disruptive effects, have been found in the Arctic populations [26].

A gender analysis of the situation in general is that men of the non-European Arctic are mostly Indigenous and rural who either engage into existing economic activities (natural resources extraction, timber industry, and reindeer herding) which implies a non-sedentary way of living and commuting between work place and community settlement or they stay unemployed [27]. Volunteered or forced unemployment in remote settlements leads to the situation of higher alcohol consumption rates and higher suicide rates, especially among men of Indigenous background. The non-sedentary way of living for employed men means that they are living in closer contact with wildlife. “Traditional” food preparation quite often avoids heat treatment. That is why the information on zoonotic diseases and toxic chemicals and other contaminants, which are immune-suppressive, is important as both animals and humans in the Arctic areas become more vulnerable to the novel diseases. In this context men in the Arctic turn to be more vulnerable with respect to their health in comparison to women with the sedentary way of living. Men and women in rural Arctic consume local fish, game, berries, and mushrooms as part of their diet. This makes zoonotic diseases a very real threat to Arctic residents in the situation of northward influx to the Arctic of new wildlife and insect species [3,4,17] as well as contaminants both locally produced and long-range transmitted.

While reports point women of childbearing age and small children as risk groups in need of protection from contaminants accumulated in traditional food, these reports also inform that it is men who in most Indigenous societies predominantly consume traditional food and become exposed to higher health risks especially at later stages of life. Combined with other factors food security more and more becomes men’s issue in rural Arctic communities. Most of the information they get is from the news, so the next question is why and how media representations of climate and environment change are produced, negotiated, and disseminated [28]. This brings the issue of adequate data and information dissemination among the Arctic residents including other gender aspects than biological sex. The Russian case study shows that there is a clear need for more studies and data, gender-divided data included on the Arctic in general and the Russian Arctic in particular. Women and men should be informed and supported differently due to their different lifestyles, responsibilities, and access to information.

In the Arctic Council gender equality is an on-going project in the Sustainable Development Working Group (SDWG), and during the Finnish Chairmanship period in 2002 gender equality was for the first time integrated into the Arctic Council chairmanship program, and 2017–2019 it has been focused as a crosscutting theme [29,30]. Gender equality was one of the themes in the Second University of the Arctic Congress in Oulu and Helsinki September 2018. Further, during the Icelandic period, it was one of the leading themes. However, there is a need to have more thorough discussion on gender equality in the Arctic Council in all its activities and actions, and also in its six working groups, not only SDWG and its subgroups, the Arctic Human Health Expert Group (AHHEG) and the Social, Economic, and Cultural Expert Group (SECEG). There is lack of gender specific data and gender mainstreaming processes in the Arctic Council activities. Every country makes its own decisions on the focus areas for two years, and mainstreaming gender equality as one project needs funding to be as a research project. Limitations of the study are the low number of respondents, but villages in northern Russia do not have many inhabitants and the participation rate was high.

## 5. Conclusions

Bacchi’s approach shows that lack of sex-disaggregated data allows a continuation of ignoring differences between women’s and men’s life styles, needs and health maintenance practices. At the same time confusing “sex” and “gender” allows ignoring the complicated nature of the two concepts and their interplay with ethnicity, place of living, age, education, and access to power and information.

Based on our case report in relation to current knowledge on broader gender perspectives not included in this case, we conclude that a broader gender analysis would allow better understanding of the processes that take place in the local rural Arctic communities. Both sex and gender data are important as they provide local politicians and decision-makers with useful information on target groups and the policies needed. Thus, this approach will allow local improvements while still having the understanding of the global changes and their impacts on the Arctic communities.

We recommend:(1)There is a need for an umbrella network on a broader gender perspective in the Arctic Region, which gathers researchers, experts, NGOs, and politicians;(2)A broader gender equality perspective should be included in all projects, activities and decision-making in the Arctic Council dealing with human health and sustainable development;(3)More funding should be made available for a broader gender research in the Arctic.

## Figures and Tables

**Table 1 ijerph-18-00628-t001:** Data base search according to Preet et al., 2010 [5], repeated on 26 March 2019.

Search Terms	PubMed 2010 * *N* (%)	PubMed 2019 *N* (%)	Web of Science 2010 * *N* (%)	Web of Science 2019 *N* (%)
Climate change	5254 (100)	33,722 (100)	47,267 (100)	201,808 (100)
Climate change and human health	879 (17)	1048 (3.1)	504 (1.1)	2659 (1.3)
Climate change and human health and gender	0 (0)	30 (0.01)	4 (0.001)	34 (0.002)
Climate change and human health and gender and policy	0 (0)	5	0	7

* Data provided from Preet et al., 2010 [5].

## Data Availability

Not applicable.

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
