# Peer review of "Climate Change in the Arctic—The Need for a Broader Gender Perspective in Data Collection"

_ijerph, 2021, doi:10.3390/ijerph18020628_

Round 1

Reviewer 1 Report

The text is much clearer now with this revision.

Reviewer 2 Report

This article discusses climate change in the arctic, which is important. However, the manuscript requires some revisions, as indicated below:

  1. Lines 14 to 15: the methods description needs to be improved in the abstract. Currently it states “The theoretical background for gender-analysis is here described together with examples form the Russian Arctic and a literature search.” I would revise this to something more specific. Here is an example: “This study involved a literature review and used a case study design to analyze gender differences in the roles and responsibilities of men and women residing in the Arctic”.
  2. Line 17: grammatically awkward. Perhaps remove “but” and replace with “and”.
  3. Lines 23 to 25: please remove the first initial from the in-text citations as only last names are required for in-text citations.
  4. Lines 31 to 32: awkward, consider revising to “The discussion at this congressional meeting revealed that women continue to experience significant inequalities whereas men enjoyed historical advantages, such as leadership positions.”
  5. Lines 32 to 33: awkward, consider revising to: “One recommendation that came out from this congressional meeting was to introduce policies seeking gender equity, and to include more men in gender equity discussions.”
  6. Line 44: grammatically awkward to state “genders”. Please revise.
  7. Lines 72 to 73: consider revising and replace with “This study consists of a literature review and a case study.”
  8. Lines 72 to 73: what kind of literature review did you conduct? Systematic or scoping?
  9. Lines 85 to 86, re: “The literature search rendered…” you should be stating what are the exact differences between this 2019 literature review from the 2010 version.  I would also make a new table to reflect your unique findings, so that we can compare it to Table 1 (the 2010 literature review data).
  10. Line 93: replace “Case report, in brief” with “Case Study Results”
  11. Line 94: replace “In the Russian case study, a total of 51%...” with “Our respondents for the questionnaire consisted of 51% men and 49% women.” What were the % men and % women for the interviews? Did you collect any other demographic data?
  12. Line 107: The citation Syed and Ahmad, 2016 is not listed in the reference list on page 7. Please list the full citation if you decide to use it.
  13. Line 111: Grammatical error. Replace “In addition, on the community level” with “In addition, at the community level”
  14. Line 156: you already indicated the % of men and women in line 94, so why are you discussing “other gender”? I feel that this is conflating the meaning of sex/gender. I would delete this line completely.
  15. Lines 182 to 183: I would avoid categorizing chronic diseases as “Western”. They are also a problem in the global south, for example in South Asia. Perhaps indicate “there is an increase in chronic disease, such as…”
  16. Line 183: please remove the first initial “A.” from the references, you only need to list the last name for the in-text citation. Please keep the first initial in the reference list at the end of the document, that is okay.
  17. Lines 191 to 192: see comment above regarding citing in-text.
  18. Line 244: Before the conclusion section, I feel that you need to comment on a few limitations of your study. Right now the limitations are not clear.
  19. Line 324: there is a spelling and spacing error, it should be “Syed, I.” not “Syid.I.”

If you decide to incorporate these suggestions, please upload a manuscript that contains tracked changes or other method to highlight revisions.  Thank you for the opportunity to review this work.

Round 2

Reviewer 2 Report

This article discusses climate change in the arctic, which is important. The authors have revised the manuscript per the reviewer’s suggestions.  I recommend the manuscript should be accepted for publication with some minor revisions. Below are some final recommendations:

  1. Line 5 is missing the close parenthesis after 1 e.g. it should be 1)
  2. Line 6 has an extra closed parenthesis after 2 e.g. 2) )
  3. Line 99 – please delete “(49% of women and 51% of men)” as this repeats information from Line 95 unnecessarily.
  4. Line 239 to 240. Please note that paragraphs are normally 3 to 5 sentences.  This sentence seems to be a paragraph on its own.  Please either expand upon the limitations or merge this sentence with the previous paragraph so that the sentence does not stand alone as a single-sentence paragraph.
  5. Line 240 – please insert a space between “participationrate” so that it is written as “participation rate”.

Thank you for the opportunity to review this work.

Author Response

This manuscript is a resubmission of an earlier submission. The following is a list of the peer review reports and author responses from that submission.

Round 1

Reviewer 1 Report

I can see the potential of the topic of this paper and of the approach, but as it is, I wouldn’t recommend publication. The structure and details of the research are lacking a clear organization, the goal of the paper is not clearly stated to the degree that the reader can only figure out the details of the research (questionnaires used with three case of studies). Some examples of the problems found:

The paper states: “Statistics is the core of monitoring and for a sustainable development anywhere; these statistics should be gender divided as men and women to a certain extent have different livelihoods and thus are exposed to different health hazards”.  This statement, that seems to be the main justification of relevance of the paper, is ungrounded (not explained nor referenced). What differences do you refer to? Of course, later in the discussion this is addressed. But it is too late to leave this for the discussion, since this is the justification of the importance of the topic addressed and thus a proper development of it should be at the beginning of the paper.

Most importantly, the goal of the paper is not clearly stated, if stated at all. It is mentioned a global need: to stress the importance of using gendered statistics as climate change in the Arctic affects women and men, girls and boys differently. But there is no clear introduction of what is the specific research goal of the paper about.

The methods used are only quickly and confusingly presented in a succinct sentence: “Data from a questionnaire with a gender approach on the attitudes to climate change in the Russian Arctic as a case report illustrates the usefulness. Case reports from Sweden and Finland show further the value of sex-disaggregated data and gender analysis in other settings in the Arctic.”

The results section should be left for the results of the paper. However, here the first part of the Results section is about a search conducted on a database to look for previous studies on the topic. This is related to the literature review or can be presented as a justification of the lack of research on the topic, but these are not results as such, just a search on literature and a very basic one.

The cases in the results section are very unclearly presented. First, we have a section called “Case reports”, in plural, where it seems only one case, from Russia, is reported. Then we have another section titled “Examples from other settings in the Artic”, that it starts with “These case reports from two other countries” without any proper introduction of what are “these case reports”. We then confirm that these are Sweden and Finland, but the cases must be clearly presented at the very beginning of the Results section.

Because neither the aim of the paper nor the method (what type of questionnaires are used and to what purpose) are clearly stated, the reader has severe difficulties to figure out what relevance the results from the three case of studies have for climate change statistics. What is the connection between women been more mistreated than mean and climate change statistics? I am convinced there is a connection, but the authors should develop it in advance of producing any results and any discussion.

In the discussion section there are a few subsections that show again a lack of good structure of the text. There is no subheading for the first part of the discussion and then there are two different level subheadings.

Furthermore, the discussion goes far beyond the results of the research presented. The conclusions also go far beyond the results of the research presented. Overall the authors seem to have a number of interesting ideas to tell and they use the discussion and conclusion to comment them, with disregard of what should be the core of the paper, their own results.

I would suggest the authors to read the papers by for instance sociologist Kari Norgaard as a reference of how to write and structure a paper in a related subject.

Reviewer 2 Report

This article discusses climate change in the arctic, which is important. However, the manuscript requires some revisions, as indicated below:

  1. The title seems awkward e.g. “in all data collections” should be “in all types of data collection”?
  2. Line 11 of the abstract indicates that climate change affects human diversity; but this is not clear from the manuscript. Please advise about this.
  3. Line 13 of the manuscript discusses a gender analysis. Given that the paper also touches upon the issue of “human health, and wellbeing” it might be worthwhile to cite Syed (2019) who discusses gender and health disparities. Full citation: Syed, I. (2019).  In Biomedicine, Thin Is Still In: Obesity Surveillance Among Racialized, (Im)migrant, and Female Bodies.  Societies.  9(3): 59-72. DOI: https://doi.org/10.3390/soc9030059
  1. Line 15 discusses “results from a study” but does not indicate if it is a mixed methods study or a qualitative study. Please expand.
  2. Line 51 Results – usually there is a methods section before results.
  3. Lines 84 to 86 – Could you please provide a quote to substantiate this claim?
  4. Lines 91 to 92 – It might be worthwhile to connect this finding to Syed and Ahmad’s (2016) work indicating that women often do this dual work paid/unpaid economic and domestic work. Full citation: Syed, I., and Ahmad, F. (2016). A Scoping Literature Review of Work-Related Musculoskeletal Disorders Among South Asian Immigrant Women in Canada.Journal of Global Health. 6(2):28-34. Available from: https://issuu.com/ghjournal/docs/spring_2016-3/1
  1. Line 98: it is awkward to start a sentence with “And” in an academic manuscript. Please revise.
  2. Lines 97-98: “more” than what? More than women? Urbanized communities?
  3. Line 112: “This is as shown still an” is very awkward. Please revise.
  4. Line 113: should be “human rights dimension” instead of “human right dimension”.
  5. Line 119 is missing a comma after Stockholm
  6. Lines 116-118: paragraphs are normally between 3 to 5 sentences. Please either expand or combine consecutive paragraphs.
  7. Line 158 seems grammatically awkward: please remove “on gender” and replace with “about gender” or “related to gender”.
  8. Lines 147 to 164: although this study about university students is important, your focus in the preceding paragraph was about loneliness, self-rated health, so it may be relevant to continue the discussion about human health research and in healthcare analysis. It may be worthwhile to cite the work of Syed (2019) who discuses a gender-based analysis in health research and healthcare settings e.g. physician prescribing practices of tranquilizers, women’s reproductive rights e.g. cesarean sections, and biomedical research e.g. using 160kg men in clinical trials. Syed, I. (2019).  In Biomedicine, Thin Is Still In: Obesity Surveillance Among Racialized, (Im)migrant, and Female Bodies.    9(3): 59-72. I would also recommend: Bourgeault, I.L.; Benoit, C.; Davis-Floyd, R. Reconceiving Midwifery; McGill-Queen’s University Press: Montreal, QC, Canada, 2004.
  9. Line 185: Why deliberately skip feminist epistemology?
  10. Line 329: please remove “In” from “In Conclusion”

Please upload the revised manuscript with a tracked changes version so that the reviewer may see the changes clearly.